# Cellular birthdate predicts laminar and regional cholinergic projection topography in the forebrain

**Kathryn C Allaway[1,2,3], William Muñoz[1,4], Robin Tremblay[1], Mia Sherer[2,3,5], Jacob Herron[2,3,5], Bernardo Rudy[1], Robert Machold[1]\*, Gordon Fishell[2,3]\***

[1]Neuroscience Institute, New York University, New York, United States;
[2]Department of Neurobiology, Harvard Medical School, Boston, United States;
[3]Stanley Center for Psychiatric Research, Broad Institute, Cambridge, United States;
[4]Department of Neurosurgery, Massachusetts General Hospital and Harvard Medical School, Boston, United States; [5]Northeastern University, Boston, United States

**Abstract** The basal forebrain cholinergic system projects broadly throughout the cortex and constitutes a critical source of neuromodulation for arousal and attention. Traditionally, this system was thought to function diffusely. However, recent studies have revealed a high degree of spatiotemporal specificity in cholinergic signaling. How the organization of cholinergic afferents confers this level of precision remains unknown. Here, using intersectional genetic fate mapping, we demonstrate that cholinergic fibers within the mouse cortex exhibit remarkable laminar and regional specificity and that this is organized in accordance with cellular birthdate. Strikingly, birthdated cholinergic projections within the cortex follow an inside-out pattern of innervation. While early born cholinergic populations target deep layers, late born ones innervate superficial laminae. We also find that birthdate predicts cholinergic innervation patterns within the amygdala, hippocampus, and prefrontal cortex. Our work reveals previously unappreciated specificity within the cholinergic system and the developmental logic by which these circuits are assembled.

**\*For correspondence:**
robert.machold@nyulangone.org (RM);
gordon_fishell@hms.harvard.edu (GF)

**Competing interests:** The authors declare that no competing interests exist.

## Introduction

Acetylcholine (ACh) plays an essential role in modulating attention, motivation, and learning within neocortical, hippocampal, and subcortical circuits (*Ballinger et al., 2016*; *Luchicchi et al., 2014*; *Picciotto et al., 2012*). The primary source of ACh within these structures are the projection neurons located in several nuclei throughout the basal forebrain. Historically, these neurons were thought to be relatively indiscriminate sources of ACh, releasing it slowly and diffusely throughout the cortex to mediate widespread circuit activation (*Sarter et al., 2009*). Nonetheless, considerable in vitro work has implied a multitude of layer-, cell type-, and synapse-specific cholinergic effects (reviewed in *Muñoz and Rudy, 2014*). Recent in vivo findings have revealed an even higher level of spatiotemporal coordination (*Froemke et al., 2007*; *Muñoz et al., 2017*). These observations hint that the ACh system is composed of distinct cell types that target specific layers and circuit elements of the cortex in a temporally precise manner.

Classically, the basal forebrain cholinergic neurons (BFCNs) have been divided into four groups based upon cell body location – Ch1 (medial septum), Ch2 (vertical diagonal band), Ch3 (horizontal diagonal band), and Ch4 (substantia innominata and nucleus basalis) (*Mesulam et al., 1983*). These broad anatomical divisions roughly reflect their innervation of different brain structures (i.e. neocortex, hippocampus, and amygdala). Further efforts delineated a more refined topography. Early work suggested that rostrally located cells project to medial cortical areas and that caudal cells project

laterally (*Baskerville et al., 1993*; *Saper, 1984*). This work was primarily carried out using antero-grade and retrograde tracers which allowed the broad spatial topography to be determined, but could not distinguish between cholinergic and noncholinergic cells. More recently, genetic tools have allowed for the sparse, selective targeting of cholinergic cell types but failed to reveal their overall topographic and anatomical organization (*Li et al., 2018*; *Wu et al., 2014*; *Zaborszky et al., 2015*). Here, by using a hybrid genetic/retrograde labeling strategy, we demonstrate the existence of cholinergic neurons with specific projections to deep, middle, and superficial layers of the mouse somatosensory cortex.

We further sought to understand how this organization in the cholinergic system arises during development. It is known that all cholinergic neurons in the forebrain originate within the Nkx2.1+ proliferative region of the ventral embryonic telencephalon (*Marin et al., 2000*; *Patel et al., 2012*; *Xu et al., 2008*). The diversity of other cell types arising from this region is generated by both spatially defined progenitor pools (*Gelman et al., 2009*; *Nóbrega-Pereira et al., 2010*; *Wonders et al., 2008*), as well as temporal shifts in the neuronal subtype produced (*Inan et al., 2012*; *Miyoshi et al., 2007*). Here, we investigated the developmental origins of cholinergic projection neuron topography. We were surprised to find that the temporal, but not the spatial, organization of the progenitors predicted the organization of the mature cholinergic projections. Furthermore, we found that the axons of cholinergic neurons born at different embryonic timepoints take distinct pathways to reach their projection targets. Together, these results illustrate that the temporal origins of the BFCNs predict the precise organization of their cortical and subcortical axonal topographies.

## Results

### Layer-specific cholinergic projections in the mature somatosensory cortex

To explore specificity of cholinergic axonal arborizations, we used a hybrid genetic/viral strategy to label small subpopulations of cholinergic neurons innervating the mature cortex. To that end, we generated a mouse line in which the *FlpO* recombinase was targeted to the choline acetyltransferase locus ($Chat^{FlpO}$). When used in the context of an *Flp*-dependent reporter, this enables the specific labeing of cholinergic neurons. This mouse line exhibits the expected selectivity of *FlpO* expression within cholinergic neurons when crossed to a pan-ventral ($Dlx6a^{Cre}$) line and visualized using an intersectional reporter ($Rosa26^{Ai65}$) (*Figure 1—figure supplement 1*).

To determine whether cholinergic neurons have restricted arborizations within specific layers of the somatosensory cortex, we crossed the $Chat^{FlpO}$ line to the $Rosa26^{Ai65}$ intersectional (i.e. *Cre* and *Flp* dependent) reporter allele and injected these animals with a *Cre*-expressing type 2 canine adenovirus (*CAV-2::Cre*). This virus specifically infects axon terminals (*Ekstrand et al., 2014*; *Junyent and Kremer, 2015*), resulting in *Cre* expression only in cells that innervate the injected region. By restricting this virus to either the superficial or deep layers of the primary somatosensory cortex of $Chat^{FlpO}$; $Rosa26^{Ai65}$ mice (*Figure 1A*), we were able to selectively label cholinergic neurons with axons projecting to specific cortical layers. The fluorescent reporter fills the entire axonal arborization of labeled cholinergic neurons and thus allowed us to determine the extent of arborization of individual cells across laminae. Using this method, we observed that the majority of projections were restricted to the injected layers. This indicates that most cholinergic neurons predominantly arborize within specific cortical laminae (*Figure 1B,C*). In particular, deep layer injections largely showed projections restricted to layers 5 and 6 (L5-6) of the somatosensory cortex, whereas superficial injections primarily showed projections within layers 1 through 4 (L1-4), with some showing remarkable specificity for L1.

The layer specificity of cholinergic projections was further confirmed by two high-resolution, single-cell reconstructions of the axonal arbors of individually labeled cholinergic neurons– one targeting L1 of the somatosensory cortex, and another targeting L5-6 (*Figure 1G,H*; *Figure 1—figure supplements 2* and *3*). The soma of the L1-targeting cell was located in the rostral substantia innominata, sending its axon rostrally then dorsally through the medial septum, ultimately entering the cortex through L1. Despite the fact that this cell had arborizations extending from the caudal motor cortex to the rostral visual cortex, it was almost entirely restricted to layer 1. Conversely, the soma of the L5/6-targeting cell was located in the nucleus basalis, near the globus pallidus. Its axon

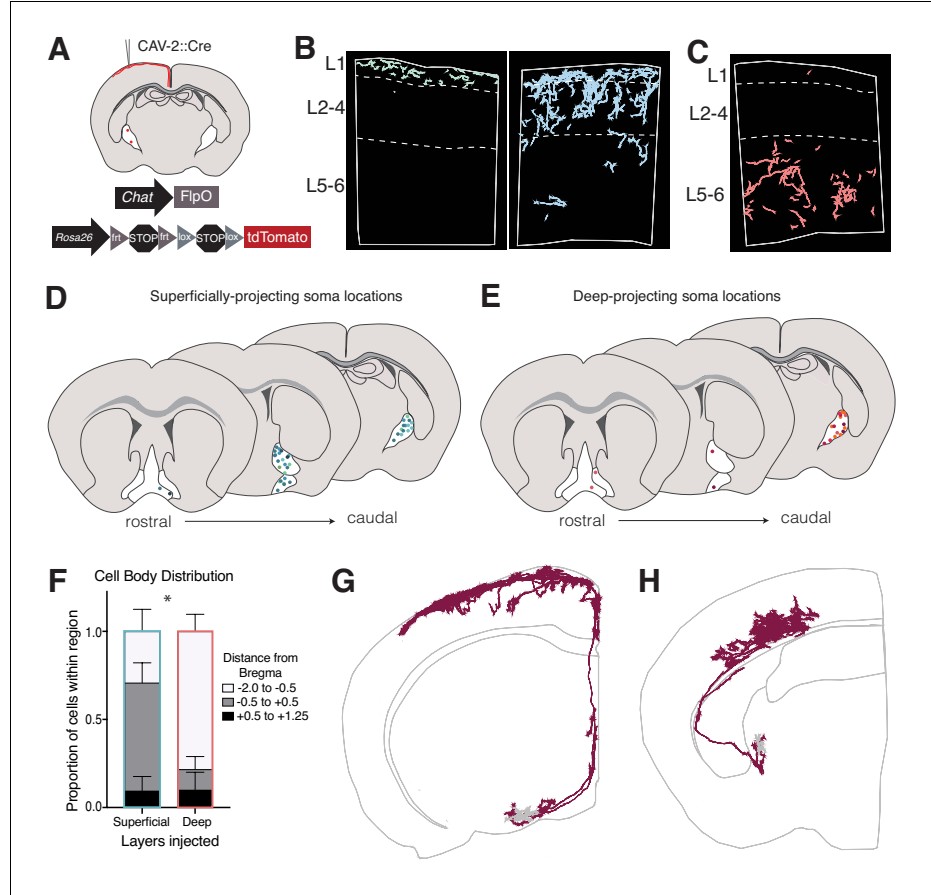

**Figure 1.** Retrograde labeling of layer-specific cholinergic projections in the somatosensory cortex. See also *Figure 1—figure supplements 1*, *2* and *3*. (A) Genetic and viral strategy for specific labeling of cholinergic efferents to superficial or deep layers of the somatosensory cortex. (B) Reconstructed fibers within representative sections of the somatosensory cortex from superficial layer injections, some of which were remarkably L1-specific (left) and some which exhibited fibers in L1-4 (right). (C) Reconstructed fibers within representative sections from a deep layer injection. (D and E) Cell body locations within the basal forebrain of cholinergic neurons labeled in superficial- (D) or deep- (E) layer injections. Shades of colors within (D) and (E) represent individual animals (superficial injections n = 6, deep injections n = 5). (F) Proportions of labeled cell bodies located across the rostral to caudal axis in the basal forebrain. Proportions were calculated for each injection individually then averaged, error bars show SEM. Two-way ANOVA $F(2,18)$ = 13.63, p=0.0037; Sidak's multiple comparisons test shows the proportion of cells located between Bregma −0.5 and +0.5 is significantly different between groups (p=0.0466). (G) Complete reconstruction of a cholinergic neuron with the majority of its axonal arborizations in L1. (H) Complete reconstruction of a cholinergic neuron with restricted projections in L5-6. Reconstructions are shown on a representative hemisphere from a coronal section. Red fibers indicate axonal projections, gray fibers indicate dendrites.

The online version of this article includes the following figure supplement(s) for figure 1:

**Figure supplement 1.** Precise and efficient recombination mediated by the *Chat*-ires-Flpo driver.
**Figure supplement 2.** Complete reconstruction of a L1-targeting cholinergic projection neuron.
**Figure supplement 3.** Complete reconstruction of a L5/6-targeting cholinergic projection neuron.

traversed through the striatum to enter the external capsule before entering the somatosensory cortex from L6. Unlike the L1-targeting cell, the arborizations of the L5/6-targeting cell were restricted to the somatosensory cortex and covered a much more limited rostral to caudal territory.

Moreover, we mapped the soma locations of the labeled cholinergic neurons within the basal forebrain in order to gain insight into whether cell body localization is related to the layer topography of their axonal arborizations (*Figure 1D,E,F*). We found that both deep- and superficial-targeting cholinergic neurons can reside within one of several nuclei across the rostral-caudal axis of the

basal forebrain, including the vertical diagonal band (vDB), horizontal diagonal band (hDB), substantia innominata (SI), and nucleus basalis (NB). We did note, however, that there is a bias for deep layer-projecting cells to be predominantly located in more caudal structures (i.e. Bregma −2.0 to −0.5, corresponding to the NB and caudal SI), while superficially projecting cells are enriched in more rostral BF structures (Bregma −0.5 to +0.5, corresponding to the hDB and rostral SI) (ANOVA, $F(2,18) = 13.63$, p=0.0037, see *Supplementary file 1* for ANOVA summary tables).

## Spatial embryonic origin within the MGE or POA does not predict adult projection patterns of cholinergic neurons

We then wanted to determine the developmental origins of these layer-specific cholinergic neurons, both to understand how this layer specificity emerges and to identify genetic tools that could be used to target these neurons for further investigation. We first asked whether layer-specific cholinergic neurons arise from distinct spatial locations within the ventral embryonic telencephalon. Embryonic progenitor zone origin has been shown to predict subtype identity within other forebrain lineages (*Bandler et al., 2017*; *Lim et al., 2018*), as well as projection neuron populations with distinct targets in the hindbrain (*Jensen et al., 2008*; *Robertson et al., 2013*; *Robertson et al., 2016*).

Previous work has shown that all BFCNs arise from the Nkx2.1+ domain within the ventral telencephalon, which encompasses both the medial ganglionic eminence (MGE) and preoptic area (POA). These two regions have distinct patterns of gene expression (*Flames et al., 2007*; *Hansen et al., 2013*) and have been shown to give rise to discrete populations of neurons (*Flandin et al., 2010*; *Gelman et al., 2011*; *Gelman et al., 2009*). In order to fate map BFCNs arising from either the MGE or POA, we crossed our $Chat^{FlpO}$ allele and intersectional reporter with either $Lhx6^{iCre}$, marking the MGE, or $Shh^{Cre}$, marking the POA, and examined the labeled cells in P30 mice (*Figure 2A*).

These fate mapping experiments revealed that both MGE-derived ($Lhx6$ lineages) and POA-derived ($Shh$ lineages) include both deep- and superficially projecting BFCNs, with no statistically significant difference in layer projection density between the two populations (*Figure 2D,E*). The labeled BFCNs from both the MGE and POA had cell bodies distributed throughout all nuclei of the basal forebrain, although we found that the MGE-derived population gave rise to a slightly higher proportion of cells within the medial septum (Bregma +0.5 to +1.25, p=0.0018) (*Figure 2B,C*). Despite this, the two populations had comparable projection densities across other cholinergic target regions, including the hippocampus, a major target of the medial septum (*Figure 2F*). These results led us to conclude that spatial origin from the MGE versus POA is not a major source of cholinergic neuron diversity with regards to layer specificity, general projection target, or cell body location.

## Cellular birthdate predicts layer-specific cholinergic projection topography

An alternative source of neuronal diversity during development is cellular birthdate - that is, the timing of a neuronal progenitor cell becoming terminally postmitotic. This principle is evident in the ordering of pyramidal neurons (*Angevine and Sidman, 1961*) and interneurons (*Inan et al., 2012*; *Miyoshi et al., 2007*) within cortical laminae. In order to determine whether layer-specific cholinergic projections correspond with neuronal birthdate, we again utilized the $Chat^{FlpO}$ and $Rosa26^{Ai65}$ alleles, this time in combination with an $Ascl1^{CreER}$ driver (i.e. $Chat^{FlpO}$; $Ascl1^{CreER}$; $Rosa26^{Ai65}$). $Ascl1$ is a proneural gene broadly but transiently expressed in newly born neurons within the ventral embryonic telencephalon (*Casarosa et al., 1999*). The expression trajectory of $Ascl1$ is closely tied to cellular birthdate, in that its expression peaks as MGE-, septal, and POA-derived progenitors exit the cell cycle, after which it is rapidly downregulated. In the context of the $Chat^{FlpO}$; $Ascl1^{CreER}$; $Rosa26^{Ai65}$ embryos, administration of tamoxifen at a given embryonic age allows the birthdate of cholinergic cells to be captured (*Battiste et al., 2007*; *Kim et al., 2008*; *Kim et al., 2011*). Cholinergic cells exiting the cell cycle immediately following tamoxifen administration (12–24 hr window) thereby express the tdTomato reporter permanently.

In order to capture the timeframe during which most cholinergic neurons are generated, we administered tamoxifen at E10, E11, E12, or E13, and then harvested the brains at P30 to examine the axonal projection topography of labeled cholinergic neurons (*Figure 3A,B*). Neurons projecting to the somatosensory cortex are generally born between E10-E12, while those labeled at E13 do not

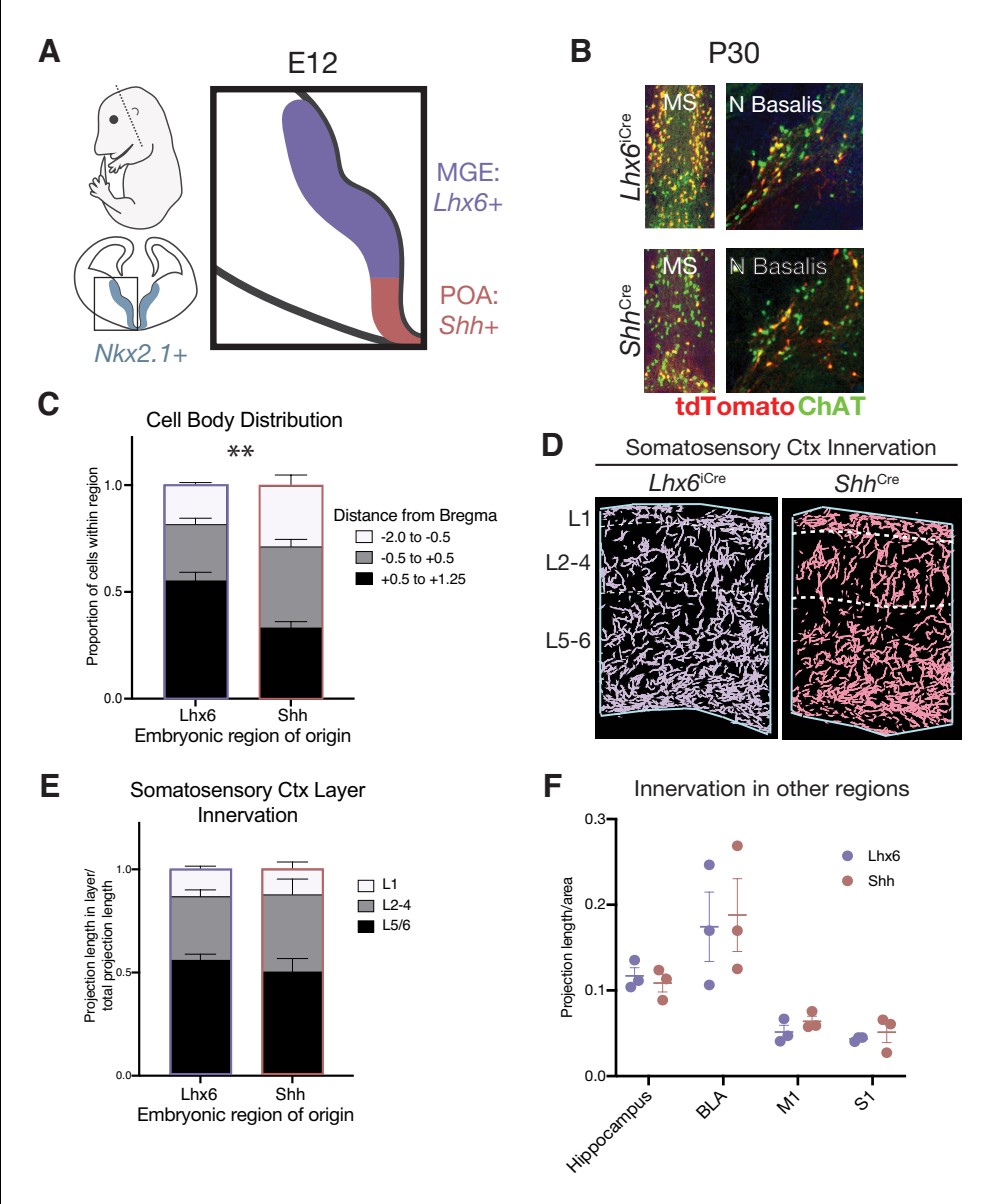

**Figure 2.** Spatial embryonic origin does not predict projection topography of cholinergic neurons. (**A**) Genetic strategy for fate mapping cells from the medial ganglionic eminence (MGE) or preoptic area (POA) based on differential gene expression. (**B**) Fate mapped cells in the P30 medial septum (MS) and nucleus basalis (N Basalis) from $Chat^{FlpO};Lhx6^{iCre};Rosa26^{Ai65}$ or $Chat^{FlpO};Shh^{Cre};Rosa26^{Ai65}$ animals stained with an anti-ChAT antibody. (**C**) Quantification of cell body distribution in the adult brain of cells originating in the MGE (Lhx6, n = 3 animals) or POA (Shh, n = 4 animals) (two-way ANOVA, $F_{(2, 10)}=9.25$, p<0.0053). (**D**) Reconstructions of axonal projections within a representative section of the P30 somatosensory cortex from fate-mapped brains. (**E**) Quantification of projection density within cortical layers for spatial fate mapped brains. Two-way ANOVA revealed no significant difference between groups ($F_{(2, 6)}=0.9034$, p=0.4311) (Lhx6 n = 3 animals, Shh n = 3 animals). (**F**) Quantification of projection density within other cholinergic target regions revealed no significant differences between groups (two-way ANOVA, $F_{(2, 6)}=0.1064$, p=0.9552). Each dot represents quantification for that region from an individual animal.

send projections to the somatosensory cortex (or most other cortical regions) with the exception of the medial prefrontal cortex (see below) (*Figure 3D*). Further analysis revealed that cholinergic neurons that become postmitotic on different days have distinct projection patterns within the primary somatosensory cortex, with early-born neurons projecting to deep layers and later born neurons

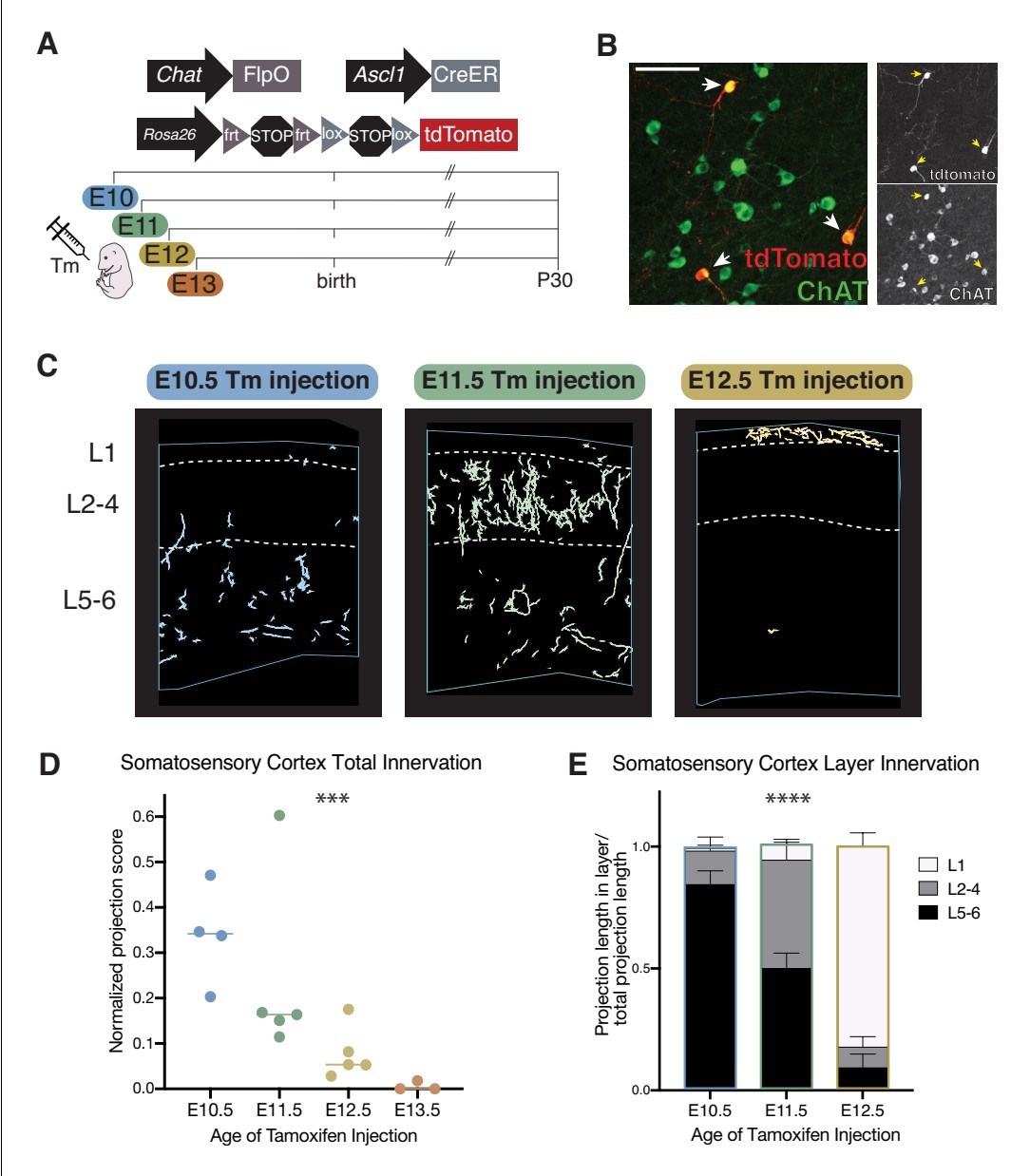

**Figure 3.** Cellular birthdate predicts layer-specific cholinergic projection topography in the primary somatosensory cortex. See also *Figure 3—figure supplement 1*. (**A**) Experimental design for tamoxifen (Tm) induction of CreER activity for neuronal birthdating. (**B**) Example of E11 tamoxifen-birthdated cholinergic neurons at P30 in the substantia innominata. Scale bar = 100 μm. (**C**) Reconstructions of birthdated cholinergic projections within representative sections of the P30 somatosensory cortex. E13 brains have little to no axonal projections within this region. (**D**) Quantification of overall somatosensory cortex innervation at P30 for cholinergic neurons born at each timepoint (one-way ANOVA ($F$(3,13) = 10.86, p=0.0008)). Each dot represents an individual animal. (**E**) Quantification of innervation to specific cortical layers for each birthdated cohort (two-way ANOVA ($F$(4, 22)=48.20, p<0.0001)).

The online version of this article includes the following figure supplement(s) for figure 3:

**Figure supplement 1.** Birthdated cholinergic neuron projections to the primary motor and primary visual cortices.

projecting progressively more superficially (ANOVA, $F$(4, 22)=48.20, p<0.0001). (*Figure 3C,E*). Axonal projections from E10-born cells were predominantly located in L5-6, while those from E11-born cells were primarily in L2-4, and those from E12-born cells were almost exclusively found in L1 of the primary somatosensory cortex. These results indicate that the layer-specific cholinergic neurons that we previously identified through viral injections become postmitotic at discrete embryonic ages.

## Birthdate predicts cholinergic projection topography throughout the forebrain

We also examined other cortical regions, including visual cortex (V1), primary motor cortex (M1), and the medial prefrontal cortex (mPFC) in order to determine whether they also contain layer-specific cholinergic projections. Importantly, we found that V1 shows similar characteristics to S1. Cholinergic neurons born at later timepoints innervate progressively more superficial layers of V1 and, like in S1, those born at E12 predominantly project to layer 1 (ANOVA, F(4, 24)=12.95, p<0.0001, *Figure 3—figure supplement 1C and D*).

M1 and mPFC, conversely, show relatively weak layer-specificity overall, suggesting that the layer-specificity of cholinergic projections is more prominent in primary sensory areas compared to non-sensory areas. In M1, while E10-born cells project primarily to L5/6, those born at E11 and E12 project across layers fairly indiscriminately (ANOVA (F(4, 14)=2.514, p=0.0888), *Figure 3—figure supplement 1A and B*). Likewise, in mPFC, which is unique amongst cortical areas in receiving projections from E13 born cholinergic neurons, early born cells primarily innervate deep layers, while later born cells appear to target all layers (ANOVA F(6, 22)=5.678, p=0.0011; *Figure 4A,D,E*).

Because the birthdating method employed here also captures cholinergic neurons innervating other forebrain regions, we next examined the hippocampus and amygdala to determine if cholinergic projections within these areas also correlated with cellular birthdate. We found that hippocampally projecting cholinergic neurons are primarily born at later timepoints (E12 and E13) (hippocampus – ANOVA, $F(3,14) = 17.51$, p<0.0001, *Figure 4B,F*). Conversely, those projecting to the basolateral amygdala (BLA) are primarily born at E11 and E12 (ANOVA, $F(3,11) = 5.691$, p=0.0133). Interestingly, BFCNs born at E10 predominantly project to the central amygdala, which receives much less overall cholinergic innervation than the BLA (ANOVA, $F(3, 25)=308.5$, p<0.0001) (*Figure 4C,G,H*).

## Soma locations of birthdated cholinergic neurons

The cell bodies of cholinergic neurons born at each timepoint were found across the rostral-caudal extent of the basal forebrain (*Figure 5A,B*). We noted a general trend that early born (and deep-layer projecting) cells were biased toward the caudal-most structures, while later born (and more superficially projecting) neurons were found more rostrally. However, this difference was not statistically significant (ANOVA, F (6, 30)=2.103, p=0.0825). We repeated this experiment with EdU, an alternative birthdating method that labels cell nuclei, and saw a similar trend that did reach statistical significance (ANOVA, F (6, 24)=3.705, p=0.0092; Figure C, D).

## Cholinergic axons traverse distinct routes to their projection targets based on birthdate

Previous studies have described several routes that cholinergic axons take when traversing the basal forebrain to reach the cortex. These axonal pathways have been linked to projection target specificity (*Bloem et al., 2014*; *Eckenstein et al., 1988*; *Saper, 1984*). In order to determine whether cholinergic neurons born at different times preferentially utilize specific axonal projection routes, we quantified the projections of birthdated cholinergic neurons passing through the (1) rostromedial, (2) septal, (3) rostrolateral, and (4) caudolateral pathways (*Figure 6A*). We found that E10 born cholinergic neurons preferentially travel via the caudolateral pathway to reach deep cortical layers (ANOVA, ($F(3,8 = 5.799)$, p=0.0209)) (*Figure 6B*). Those born at E11, which primarily project to layers 2–4, were found to have most fibers in the rostrolateral and septal pathways, suggesting these axons reach the cortex via either route (ANOVA, $F(3,12) = 20.14$, p<0.0001). E12 and E13 born neurons, conversely, primarily travel medially through the septal pathway (E12) (ANOVA, $F(3,8) = 13.63$, p=0.0016) or the rostro-medial pathway (E13) (ANOVA, $F(3,8) = 5.599$, p=0.0230). Together, these results suggest that cholinergic neurons projecting to deep layers of the somatosensory cortex reach their targets by traversing through the lateral route, while the L1-specific cholinergic neurons reach it via the septal pathway. E13 born neurons do not target most of the cortex outside of the mPFC, and reach this target via the rostro-medial route.

To investigate this further, we again utilized the CAV-2::*Cre* retrograde virus, this time injecting it directly into either the septal or caudolateral pathway of *Chat^{FlpO}; Rosa26^{Ai65}* mice. Quantification of cholinergic projections within the somatosensory cortex labeled using this method confirmed that

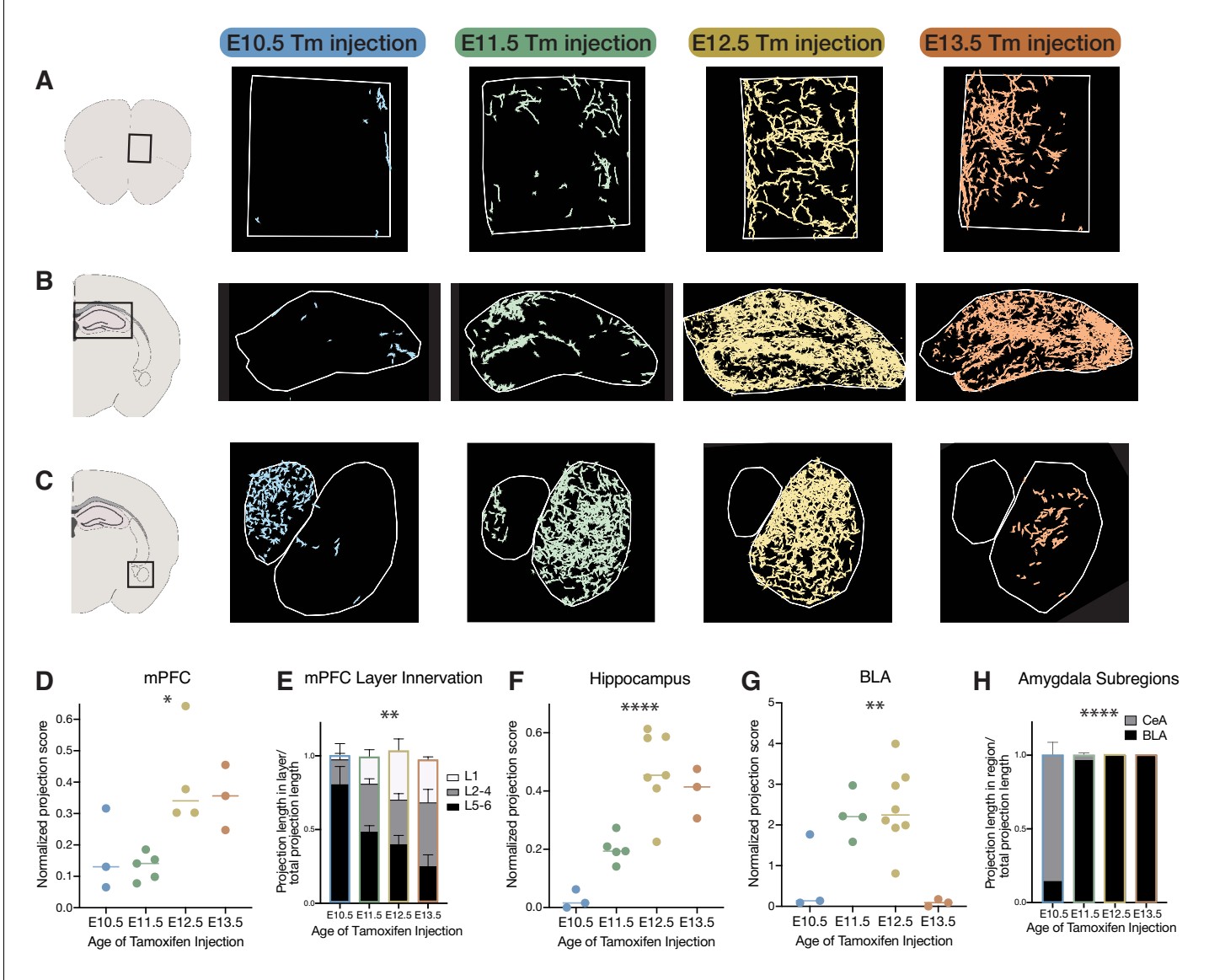

**Figure 4.** Birthdate predicts cholinergic topography in the hippocampus, amygdala, and mPFC. (**A**) Examples of birthdated cholinergic neuron projections to the medial prefrontal cortex (mPFC). (**B**) Examples of birthdated cholinergic neuron projections to the hippocampus at P30. (**C**) Examples of birthdated cholinergic neuron projections to the amygdala. Right, smaller outline represents central amygdala (CeA) and left, larger outline represents basolateral amygdala (BLA). (**D**) Quantification of total projections to the mPFC from birthdated cohorts of cholinergic neurons (one-way ANOVA, $F(3,11) = 5.691$, p=0.0133). (**E**) Quantification of innervation to specific layers of mPFC for each birthdated cohort (two-way ANOVA ($F(6, 22) =5.678$), p=0.0011). (**F**) Quantification of total projections to the hippocampus from birthdated cohorts of cholinergic neurons (one-way ANOVA, $F(3,14) = 17.51$, p<0.0001). (**G**) Quantification of total projections to the amgydala from birthdated cohorts of cholinergic neurons (one-way ANOVA, $F(3,14) = 8.219$, p=0.0021). (**H**) Quantification of projections to the CeA versus BLA for each birthdated timepoint (two-way ANOVA, $F(3, 25)=308.5$, p<0.0001). Each dot in D, F, and G represents an individual animal.

axons traveling via the septal route primarily arborize within superficial layers, particularly layer 1 (*Figure 6C,E*). Conversely, those traveling via the caudolateral route are relatively restricted to deep layers (*Figure 6D,E*). Together, these results indicate that early born cholinergic neurons project their axons via the caudolateral route to innervate deep layers of somatosensory cortex, while the axons of later born cells traverse the septal route to innervate superficial layers.

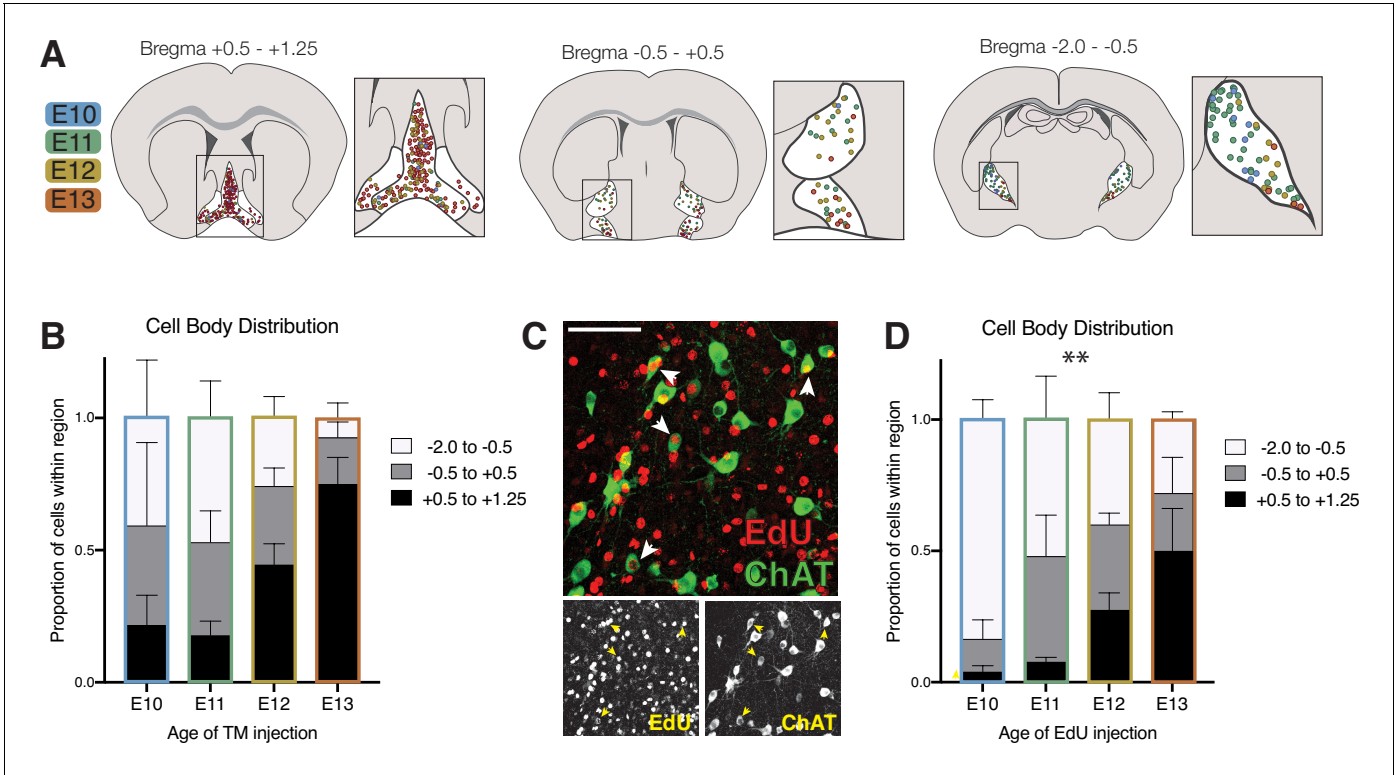

**Figure 5.** Cell body distribution of birthdated cholinergic neurons. (**A**) Cell body locations in the basal forebrain at P30 for cholinergic neurons born at each timepoint labeled by tamoxifen induction of CreER. (**B**) Quantification of cell body distribution (not significant by two-way ANOVA, $F_{(6, 30)}=2.103$, p<0.0825) (E10 n = 3; E11 n = 5; E12 n = 8; E13 n = 3). (**C**) Birthdating using the alternative 5-Ethynyl-2'-deoxyuridine (EdU) method. Example image of P30 brain in which EdU was injected at E11, with an anti-ChAT antibody stain to quantify cholinergic neurons. (**D**) Quantification of cell body distribution at P30 for cholinergic neurons birthdated with EdU (two-way ANOVA, $F_{(6, 24)}=3.705$, p=0.0092) (n = 4 for each timepoint).

## Discussion

Although originally thought of as a diffuse, nonspecific source of neuromodulation in the brain, BCFNs are now understood to function with great specificity and precision. However, the organization of this system, how it develops, and how this organization may mediate its specificity is still poorly understood. Here, we show that cholinergic neurons targeting the somatosensory cortex innervate specific layers – an organizational principle that was previously unknown and likely significantly contributes to the laminar specificity of cholinergic signaling (*Muñoz and Rudy, 2014*; *Muñoz et al., 2017*; *Obermayer et al., 2017*). Furthermore, we found that this specificity is determined by the birthdate of cholinergic neurons in the BF, with early born cholinergic neurons primarily targeting deep layers and later born neurons targeting progressively superficial layers of the somatosensory cortex. Importantly, we identified a population of cholinergic neurons whose axonal projections are almost completely restricted to layer 1. Cholinergic neuronal birthdate is also correlated with projection topography in other target regions, including other cortical areas, hippocampus, and amygdala. Finally, we found that cholinergic neurons born at different times project via distinct axonal pathways to reach their targets. Together, these findings extend our understanding of the organization of BFCNs by revealing the relationship between their developmental origins and their specific projection fields.

### Layer-specificity of cholinergic neurons

Given the distinct actions of ACh in different cortical laminae, the existence of layer-specific cholinergic neurons has been previously hypothesized. Here, using intersectional genetics, we were able to identify layer-specific populations in the somatosensory cortex for the first time. Perhaps most notably, we discovered the existence of a population of cholinergic neurons, primarily born at E12 that almost exclusively innervate layer 1. This population may explain the high density of cholinergic

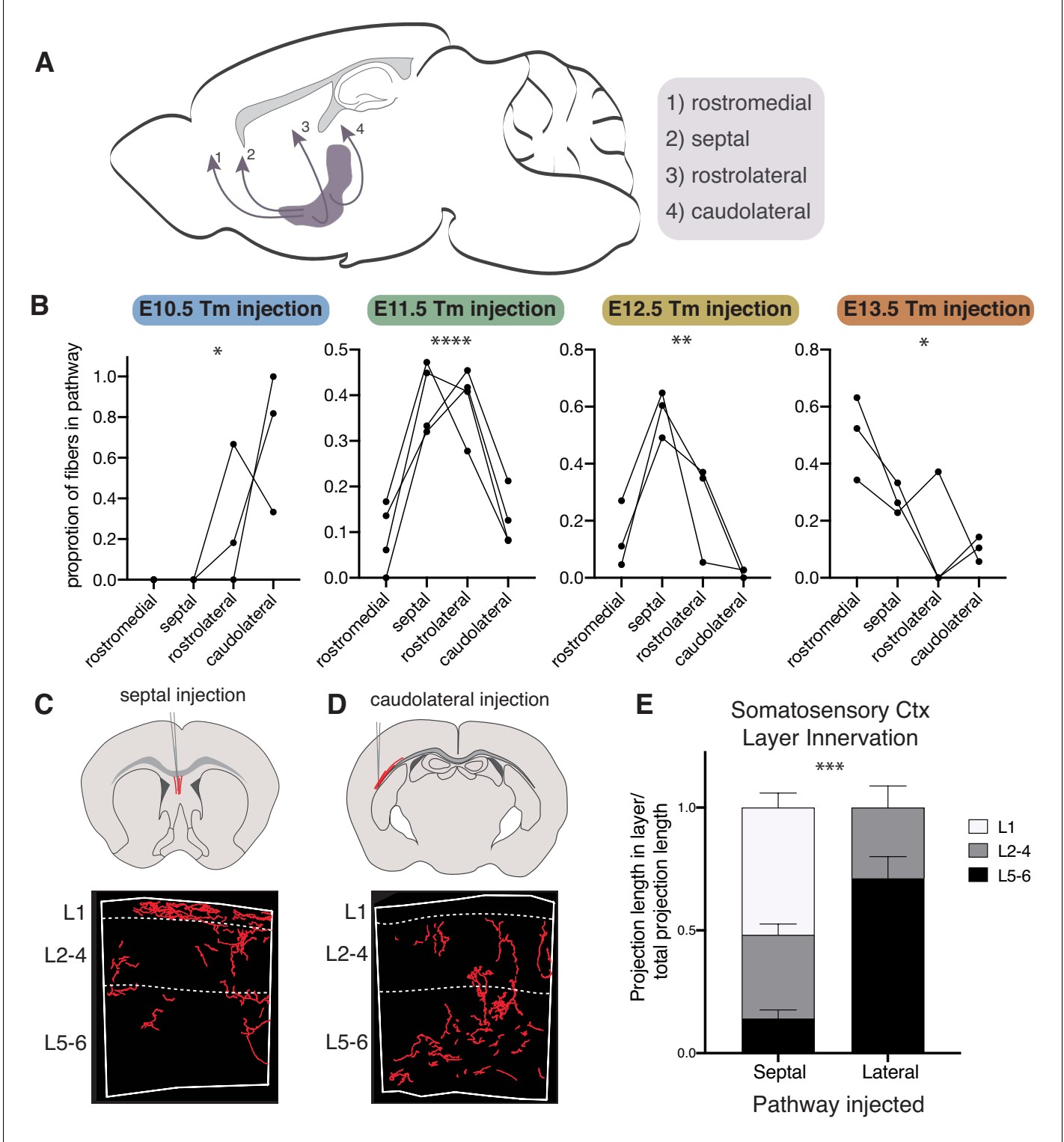

**Figure 6.** Cholinergic axons traverse distinct routes to their projection targets based on birthdate. (A) Routes cholinergic axons take to the cortex from the basal forebrain. (B) Proportion of fibers in each pathway for brains with birthdated cholinergic neurons at P30. Connected lines represent a single animal. One-way ANOVAs: E10 - $F_{(3,8)}$ = 5.799, p=0.0209; E11 - $F_{(3,12)}$ = 20.14, p<0.0001; E12 - $F_{(3,8)}$ = 13.63, p=0.0016; E13 - $F_{(3,8)}$ = 5.599, p=0.0230. (C) Septal injection schematic (top) and example of fiber reconstruction in the somatosensory cortex (bottom). (D) Caudolateral injection schematic (top) and example of fiber reconstruction in the somatosensory cortex (bottom). (E) Quantification of innervation to specific layers in septal or caudolateral pathway injected brains (n = 3 for each) (two-way ANOVA, $F_{(2, 8)}$=28.87, p=0.0003).

axons in L1 compared to other layers. Moreover, it has important functional implications for ACh in cortical processes regulated by L1, such as cross-modal signaling.

Importantly, we observed marked differences in the degree of innervation specificity within sensory (S1 and V1) and non-sensory (mPFC and M1) cortical regions. In a recent anterograde tracing study, it was shown that labeled cohorts of cholinergic neurons targeting the mPFC could be divided into two groups: one group projecting to all layers, and another projecting to deep layers only (*Bloem et al., 2014*). Our results corroborate this finding: projections from early born cholinergic neurons are relatively restricted to deeper layers, while projections from later born cells were visible across all layers of mPFC. M1 showed a similar trend, with some labeled cohorts of neurons showing deep layer projection specificity, while others more diffusely innervated the entire cortical mantle. This discrepancy between the degree of innervation specificity observed in sensory versus non-sensory areas may relate to the distinct roles of cholinergic signaling in information flow within these regions (bottom-up versus top-down, respectively).

## The role of spatial versus temporal cues in BF organization

We initially expected that cholinergic neurons targeting different layers or regions might originate from spatially distinct progenitor zones. Instead, however, we found no major differences in the contribution of cholinergic cells that emanate from the MGE or POA with regards to their cell body location or projection topography. Complementing our results, a recent study fate mapped cholinergic neurons arising from the embryonic septum, another Nkx2.1+ region. Like the MGE and POA, the septum gives rise to cholinergic neurons whose axons have no apparent specificity within the cortex (*Magno et al., 2017*). Together, these results show that these broad embryonic zones all produce heterogenous populations of cholinergic neurons that, as a group, project throughout the forebrain. However, we cannot rule out the possibility that smaller, more specific subregions of these embryonic structures could give rise to populations of cholinergic neurons with more defined projection topography.

By contrast, we discovered that cellular birthdate defines subpopulations of cholinergic neurons with distinct projection distributions. How this specificity is achieved is unclear. Are cholinergic populations specified as 'deep layer-targeting' or 'superficial layer-targeting' as soon as they become postmitotic or do these reflect temporal shifts in guidance cues that both dictate their projection trajectory and laminar specificity? Although at present prohibitively challenging, one could potentially discern between these possibilities through heterochronic transplantation of birthdated, postmitotic cholinergic neurons.

Furthermore, cholinergic projections could initially project broadly across cortical laminae and become refined to specific layers through a combination of synaptic pruning and programmed cell death. In cholinergic neurons, both these processes are regulated in part by the p75 neurotrophin receptor (*Barrett, 2000*; *Boskovic et al., 2019*; *Niewiadomska et al., 2011*). Additionally, attractant or repellant cues secreted from cortical cells could play a key role in guiding cholinergic fibers to the appropriate layers. This is a particularly compelling idea given that the 'inside-out' innervation pattern of cholinergic neurons observed here reflects the manner in which the cortex itself is generated (*Angevine and Sidman, 1961*). Further investigation into the precise signaling mechanisms within the cortex that establish and maintain the laminar specificity of cholinergic neurons will be an extremely interesting avenue for future study.

## Cholinergic neuron diversity

Our work adds a level of complexity in our understanding of cholinergic neuron diversity. Cholinergic diversity likely extends beyond differences in their axonal targeting, with subpopulations possessing distinct functional and electrophysiological properties. For example, a recently published study suggests that different basal forebrain cholinergic cell types are responsible for the two timescales of ACh release in the cortex, with one mediating slow, volume transmission and the other characterized by phasic, point-to-point release (*Laszlovszky et al., 2020*). Additionally, it has recently been shown that cholinergic neurons directly within the globus pallidus are more intimately involved in basal ganglia circuitry and have specialized firing properties when compared to their neighbors in the nucleus basalis (*Saunders et al., 2015*). In addition to the BFCNs, other potential sources of ACh exist in the forebrain. For example, cholinergic neurons in the pedunculopontine and laterodorsal tegmental

nuclei of the brainstem innervate the basal ganglia and thalamus, but do not appear to directly innervate most cortical areas (*Huerta-Ocampo et al., 2020*; *Martinez-Gonzalez et al., 2011*). In addition, a population of ChAT+interneurons are found within the cortex, although until recently it was unclear whether these cells actually release ACh. Recent work, however, has shown that a subset of these cells do appear to in some capacity release ACh (*Granger et al., 2020*; *Obermayer et al., 2019*), although their functional role has yet to be fully resolved.

The recent advances in single-cell genomics technologies should provide useful insight into the full diversity of forebrain cholinergic neurons. Linking molecular identities with function and connectivity will be a crucial step toward understanding the diversity and organization of this complex neuromodulatory system.

# Materials and methods

## Key resources table

| Reagent type (species) or resource | Designation | Source or reference | Identifiers | Additional information |
|---|---|---|---|---|
| Strain, strain background (*canine adenovirus*) | Cav-2::Cre | Plateforme de Vectorologie de Montpellier | CAV Cre | |
| Genetic reagent (*Mus musculus*) | $Chat^{Flpo}$ | This paper | N/A | Submitted to Jax, further information about generation of this line in Materials and methods below. |
| Genetic reagent (*Mus musculus*) | $CMV^{Cre}$ | Jax | RRID:IMSR_JAX:006054 | |
| Genetic reagent (*Mus musculus*) | $Ascl1^{CreER}$ | Jax | RRID:IMSR_JAX:012882 | |
| Genetic reagent (*Mus musculus*) | $Lhx6^{iCre}$ | Jax | RRID:IMSR_JAX026555 | |
| Genetic reagent (*Mus musculus*) | $Shh^{eGFP-Cre}$ | Jax | RRID:IMSR_JAX005622 | |
| Genetic reagent (*Mus musculus*) | $Rosa26^{Ai65}$ | Jax | RRID:IMSR_JAX021875 | |
| Antibody | Anti-DsRed (rabbit polyclonal) | Clontech | RRID:AB_10013483 | (1:1000) |
| Antibody | Anti-ChAT (goat polyclonal) | Millipore | RRID:AB_2079751 | (1:250) |
| Antibody | Anti-rabbit IgG Alexa Fluor 594 (donkey polyclonal) | Thermo Fisher | RRID:AB_141637 | (1:1000) |
| Antibody | Anti-goat IgG Alexa Fluor 594 (donkey polyclonal) | Thermo Fisher | RRID:AB_2534105 | (1:1000) |
| Commercial assay or kit | Click-iT EdU kit for imaging Alexa Fluor 488 | Thermo Fisher | C10337 | |
| Chemical compound, drug | 5-Ethynyl-2´-deoxyuridine (EdU) | Invitrogen | A10044 | (50 µg per 1 g body weight) |
| Software, algorithm | Neurolucida 360 | MBF Biosciences | Neurolucida 360 2019 and 2020 | |
| Other | DAPI stain | Invitrogen | D1306 | (1 µg/mL) |

## Construction of the Chat-ires-Flpo driver line

A targeting construct comprised of 5' and 3' homologous arms flanking an ires-Flpo-polyA cassette and a floxed neo cassette for positive selection was electroporated into C57BL/6 ES cells (B4). Correctly targeted ES cell clones were selected by long-range PCR and restriction mapping, and subsequently injected into recipient blastocysts to create chimeric founder mice that were then bred with *Cre* deleter mice (CMV$^{Cre}$; Jax #006054) to remove the neo cassette and obtain germline

transmission. Once established, heterozygous *Chat*-ires-Flpo (*Chat^FlpO^*) were bred with C57BL/6J mice (Jax #000664) to remove the *Cre* deleter allele, following which the *Chat^FlpO^* line was bred to homozygosity.

## Animals

All mouse colonies were maintained in accordance with the Institutional Animal Care and Use Committees of NYU School of Medicine and Harvard Medical School. In addition to *Chat^FlpO^* described above, the following mouse strains were used: Swiss Webster (Taconic Biosciences), *Ascl1^CreER^* (Jax #012882) (**Kim et al., 2011**), *Lhx6^iCre^* (Jax #026555) (**Fogarty et al., 2007**), *Shh^eGFP-Cre^* (Jax #005622) (**Harfe et al., 2004**), and *Rosa26^Ai65^* (Jax #021875) (**Madisen et al., 2015**).

## Cav-2::Cre injections

Cav-2::Cre virus was obtained from the Plateforme de Vectorologie de Montpellier. Prior to injections, mice were anesthetized using 5% isoflurane followed by maintenance on 2% isoflurane. To limit the spread of viral particles allowing for layer- or pathway-specific injections, iontophoresis (Stoelting, digital Midgard precision current source) was used for all injections at 5 µA, 7 s on/3 s off, for a total of 10 min. The following stereotaxic coordinates were used: deep layer S1 (AP: −1, ML: 3, DV: 0.85); superficial layer S1 (AP: −1, ML: 3, DV: 0.15, 25° tilt), caudolateral pathway (AP: −1.4, ML: 4, DV: −2), medial pathway (AP: 1.7, ML: 0.25, DV: 3). All injections were performed on mice aged P30-P33. Ten days following injection, mice were euthanized and perfused for analysis.

## Fate mapping spatial embryonic origins

*Chat^FlpO^* mice were crossed with either *Lhx6^iCre^* or *Shh^eGFP-Cre^* mice to produce compound *ChAT^FlpO^;Lhx6^iCre^* and *Chat^FlpO^;Shh^eGFP-Cre^* alleles. Males containing both *CatT^FlpO^* and a *Cre* allele were then crossed with female *Rosa26^Ai65^* mice and the resulting pups were genotyped to identify individuals with all three alleles. At P30, these pups were transcardially perfused with PBS and 4% paraformaldehyde. Brains were harvested, fixed overnight in 4% PFA at 4°C, and sectioned at 50 µm on a Leica VT 1200S Vibratome.

For projection and soma location quantification, sections were treated with a blocking solution of 5% normal donkey serum, 0.3% Triton-X, and PBS for 30 min at room temperature. Sections were then incubated in a primary antibody cocktail consisting of rabbit anti-DsRed (1:1000 dilution, Clontech 632496) and goat anti-ChAT (1:250 dilution, Millipore AB144P) overnight at 4°C. After PBS washes, sections were incubated in secondary antibodies (Alexa Fluor 594 donkey anti-rabbit, Alexa Fluor 488 or 647 donkey anti-goat, 1:1000 dilution) in the dark for 1 hr at room temperature, followed by additional washes in PBS, treatment with DAPI, and mounting of sections on slides.

## Birthdating with tamoxifen

*Chat^FlpO^;Ascl1^CreER^* males were crossed with *Rosa26^Ai65^* females and plugs were checked daily, with the morning that a plug was observed being considered E0.5. Pregnant dams were injected intraperitoneally with 2 mg of tamoxifen (100 µl of 20 mg/ml tamoxifen, dissolved in corn oil) between E10.5-E13.5. When pups were not delivered by noon on E19.5, pups were delivered by cesarian section and fostered. At P30, pups were transcardially perfused with PBS and 4% paraformaldehyde. Brains were harvested, fixed overnight in 4% PFA at 4°C, and sectioned at 50 µm on a Leica VT 1200S Vibratome.

For projection and soma location quantification, sections were treated with a blocking solution of 5% normal donkey serum, 0.3% Triton-X, and PBS for 30 min at room temperature. Sections were then incubated in a primary antibody cocktail consisting of rabbit anti-DsRed (1:1000 dilution, Clontech 632496) and goat anti-ChAT (1:250 dilution, Millipore AB144P) overnight at 4°C. After PBS washes, sections were incubated in secondary antibodies (Alexa Flour 594 donkey anti-rabbit, Alexa Flour 488 donkey anti-goat, 1:1000 dilution) in the dark for 1 hr at room temperature, followed by additional washes in PBS, treatment with DAPI, and mounting of sections on slides.

## Birthdating with EdU

Timed pregnant Swiss Webster females (Taconic) were injected with EdU (50 µg per 1 g body weight) between E10.5-E13.5. At P30, pups were transcardially perfused with PBS and 4%

paraformaldehyde. Brains were harvested, fixed overnight in 4% PFA at 4°C, and sectioned at 50 µm on a Leica VT 1200S Vibratome. For soma location quantification, sections were treated with a blocking solution of 5% normal donkey serum, 0.3% Triton-X, and PBS for 30 min at room temperature. Sections were then incubated in goat anti-ChAT (1:250 dilution, Millipore AB144P) overnight at 4°C. After PBS washes, sections were incubated in Alexa Flour 594 donkey anti-goat (1:1000 dilution) in the dark for 1 hr at room temperature, followed by additional washes in PBS. Sections were then treated with the Click-iT EdU kit for imaging Alexa Fluor 488 (ThermoFisher, C10337), followed by DAPI, additional PBS washes, and mounting of sections on slides.

## Soma location analysis

Temporally or spatially fate mapped brains were sectioned and stained as described above. Every eighth section from each brain was examined under a fluorescence microscope and tdTomato+ cell bodies were annotated onto a brain atlas. For figures, cell locations were approximated to their location on a representative atlas for that rostro-caudal region. Cells were summed and the proportion of cell bodies within each region per brain was calculated. ANOVA was used to determine whether significant differences between the proportions of cells within each region existed across groups, with post-hoc tests using Sidak's (spatial fate mapping) or Tukey's (temporal fate mapping) multiple comparisons tests.

## Neurolucida reconstructions and projection quantification

For whole neuron reconstructions, brains were chosen with extremely sparse viral labeling to ensure clarity assigning projections to the cell of interest. Sections containing the cell of interest were imaged on a Zeiss LSM 800. Z-stacks of images were then loaded into Neurolucida 360 (MBF Biosciences) and trees were reconstructed using the 'user guided' option with Directional Kernels. Contours of major features in the brain sections were also traced for alignment of sections to produce the final reconstruction.

For the quantification of neuronal projections given brain regions, confocal images were taken of relevant brain regions. The total projections within that region were reconstructed in Neurolucida 360 and the total projection length was quantified using Neurolucida explorer. To account for the fact that tamoxifen induction of *CreER* activity results in a variable number of neurons labeled across brains, this quantification was normalized in one of two ways. For overall projection density quantification within a given region, total projection length was normalized to the number of cells labeled in that brain to give a 'normalized projection score' (total projection length/number of cells*1000). For layer analysis, the projection within a given layer was simply quantified as a proportion of the total projection length within that cortical region.

## Pathway analysis

Cholinergic axonal trajectories in birthdated brains were quantified by counting the number of fiber segments within a given pathway (rostral, septal, rostrolateral, or caudolateral). This number was then converted to a proportion relative to the contribution of other pathways to account for variable labeling in cell number across brains. The rostral pathway was defined as fibers running vertically (i.e. parallel to the edge of the mPFC) in the superficial-most part of L1 of the medial cortex in sections rostral to the medial septum. Fibers in the septal pathway were those in the medial part of sections containing the septum that perforated the corpus callosum or ran vertically through L1 in the cingulate cortex. The rostrolateral pathway was defined as fibers running through the external capsule in sections containing the medial septum or those more rostral; the caudolateral pathway fibers were those in the external pathways in sections caudal to this.

## Sample size

No statistical method was used to determine sample sizes. Sample size for each experiment can be found in figure legends. For fate mapping experiments, mice from at least two distinct litters per timepoint were used in analysis.

## Acknowledgements

We thank the Fishell and Rudy laboratories for helpful feedback and discussion, G Pouchelon for critical reading of the manuscript, and N Yusuf and M Fernandez-Otero for technical assistance. KCA is supported by NIH NRSA F31NS103398. BR is supported by NIH P01NS074972, R01NS107257, and R01NS110079. RM is supported by NIH P01NS074972. GF is supported by NIH grants R01MH071679, R01NS081297, 5P01NS074972, and UG3MH120096, as well as support from the Simons Foundation Award 566615. The *Chat*-FlpO mouse was made in collaboration with the NYU Langone Rodent Genetic Engineering Laboratory directed by Dr. Sang Yong Kim, with partial support from P30CA016087.

## Additional information

### Funding

| Funder | Grant reference number | Author |
|---|---|---|
| National Institute of Neurological Disorders and Stroke | F31NS103398 | Kathryn C Allaway |
| National Institutes of Health | P01NS074972 | Bernardo Rudy<br>Robert Machold<br>Gordon Fishell |
| National Institutes of Health | R01NS107257 | Bernardo Rudy |
| National Institutes of Health | R01NS110079 | Bernardo Rudy |
| National Institutes of Health | R01MH071679 | Gordon Fishell |
| National Institutes of Health | R01NS081297 | Gordon Fishell |
| National Institutes of Health | UG3MH120096 | Gordon Fishell |
| Simons Foundation | 566615 | Gordon Fishell |
| National Institutes of Health | P30CA016087 | Robert Machold |
| National Institutes of Health | 5P01NS074972 | Gordon Fishell |

The funders had no role in study design, data collection and interpretation, or the decision to submit the work for publication.

### Author contributions

Kathryn C Allaway, Conceptualization, Formal analysis, Investigation, Visualization, Writing - original draft, Project administration, Writing - review and editing; William Muñoz, Conceptualization, Formal analysis, Investigation, Writing - review and editing; Robin Tremblay, Formal analysis, Investigation, Visualization; Mia Sherer, Jacob Herron, Investigation; Bernardo Rudy, Supervision, Funding acquisition, Writing - review and editing, Conceptualization; Robert Machold, Conceptualization, Resources, Supervision, Funding acquisition, Investigation, Writing - original draft, Project administration, Writing - review and editing; Gordon Fishell, Conceptualization, Supervision, Funding acquisition, Writing - original draft, Project administration, Writing - review and editing

### Author ORCIDs

Kathryn C Allaway (iD) https://orcid.org/0000-0001-5975-880X
William Muñoz (iD) http://orcid.org/0000-0002-1354-3472
Gordon Fishell (iD) https://orcid.org/0000-0002-9640-9278

### Ethics

Animal experimentation: All mouse colonies were maintained in accordance with the Institutional Animal Care and Use Committees of NYU School of Medicine (IACUC Protocol 160407) and Harvard Medical School (IACUC Protocol IS00001269).

Decision letter and Author response
Decision letter https://doi.org/10.7554/eLife.63249.sa1
Author response https://doi.org/10.7554/eLife.63249.sa2

## Additional files

### Supplementary files
• Supplementary file 1. ANOVA summary tables and post-hoc test information.

• Transparent reporting form

### Data availability
All data generated or analyzed during this study are included in the manuscript and supporting files.

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
