## [Decision Letter]

Thank you for submitting your article "Cellular birthdate predicts laminar and regional cholinergic projection topography in the forebrain" for consideration by *eLife*. Your article has been reviewed by two peer reviewers, and the evaluation has been overseen by Sacha Nelson as the Reviewing Editor and Huda Zoghbi as the Senior Editor.

The reviewers have largely concurring views and the Reviewing Editor has drafted this decision to help you prepare a revised submission.

The revisions requested below only address clarity and presentation.

Summary:

The authors use intersectional genetic fate mapping to reveal previously unknown specificity in the development and projection patterns of central cholinergic neurons of the mammalian basal forebrain. This fills an important gap in our understanding of this modulatory system long known to be critical to forebrain function.

Essential revisions:

See the suggestions for textual revisions below.

Reviewer #1:

This is a simple but elegant manuscript that examines the projection rules for forebrain cholinergic neurons, namely, the relationship between their origin during development and their projection. The authors used intersectional dual-recombinase-dependent reporter activation to label cholinergic neurons (expressing ChAT-FlpO) that project either to the superficial or deep layers of the primary somatosensory cortex by restricting Cre expression to those layers using viral injections. This showed a layer-restricted projection. They then moved on the determine whether projection was dependent on spatial embryonic origin, using Lhx6Cre or ShhCre, and suggested that the origin does not predict the projection pattern. Using a tamoxifen-inducible Cre line (Ascl1CreER) the authors find that lower layers are innervated primarily by earlier-born cholinergic neurons. Birthdate also predicted the topographic projection in other regions of the forebrain including the amygdala and the hippocampus. And finally, by mapping the position and projection paths of birthdated neurons the authors find rostro-caudal differences in their localization and the axonal projection routes.

I have only minor concerns that can be addressed with minor revisions:

In Figure 2 and in the Results and Discussion the authors conclude that “Spatial embryonic origin does not predict projection topography of cholinergic neurons”. I would say that this is an overstatement given the tools that the authors have used. Shh-Cre labels indeed POA-derived neurons but Lhx6-Cre labels all the other populations of cholinergic neurons, hence, delineates a broad embryonic origin. The conclusion from the findings here should be that “POA and MGE/septal-derived cholinergic neurons project equally to different forebrain areas.”. This leaves open the possibility that smaller spatially-restricted populations of neurons may have more restricted topographic projections.

In the Discussion the authors should briefly talk about the possibility that cell death may contribute e.g. to the final laminar specificity that is observed. Also, a discussion on the timing of development of the target neurons or target areas relative to the cholinergic neurons would be useful. Are early-born targets innervated by early-born cholinergic neurons? This seems to be largely the case from the data presented.

Reviewer #2:

This is a lovely study. It demonstrates several new findings that are highly significant for understanding the organization of the cholinergic projections within the forebrain.

The authors convincingly show that cholinergic neurons have quite specific laminar termination zones in sensory cortex, that neurons projecting to deeper layers are born earlier and those projecting more superficially are born later. In addition, there is the very important finding that the structure where the cells originate is generally not clearly related to their laminar termination zones. The system in non-sensory areas seems to be slightly distinct but also with distinct distributions of targeting to structures based more loosely on birthdate.

The methods are elegant and definitive. I have essentially no substantive criticisms of the manuscript as it is.

I think there are a couple of remaining questions (outside the scope of this study but sensible to consider in the future) perhaps worth discussing. It seems that it isn't known from this study is whether the laminar projection patterns are initially specific or become so upon later refinement (as happens with many projections). It seems possible that cells projecting to deep layers in a limited r/v domain may initially project more widely in the radial domain and then be refined to be in deep layers later? Similarly, the superficially projecting neurons could also be more radially distributed in their termination initially and then refine. This latter possibility for the Layer 1 projecting cells seems perhaps less likely since they project within Layer 1 for a long way and stay in that lamina – but then perhaps there are selective cues for these cells to stay in this lamina that are made by either selective interneurons or CR cells. Again, both of these possibilities seem like interesting follow-up questions.

---

## [Author Response]

Reviewer #1:[…] I have only minor concerns that can be addressed with minor revisions:In Figure 2 and in the Results and Discussion the authors conclude that “Spatial embryonic origin does not predict projection topography of cholinergic neurons”. I would say that this is an overstatement given the tools that the authors have used. Shh-Cre labels indeed POA-derived neurons but Lhx6-Cre labels all the other populations of cholinergic neurons, hence, delineates a broad embryonic origin. The conclusion from the findings here should be that “POA and MGE/septal-derived cholinergic neurons project equally to different forebrain areas.”. This leaves open the possibility that smaller spatially-restricted populations of neurons may have more restricted topographic projections.In the Discussion the authors should briefly talk about the possibility that cell death may contribute e.g. to the final laminar specificity that is observed. Also, a discussion on the timing of development of the target neurons or target areas relative to the cholinergic neurons would be useful. Are early-born targets innervated by early-born cholinergic neurons? This seems to be largely the case from the data presented.Reviewer #2:[…] I think there are a couple of remaining questions (outside the scope of this study but sensible to consider in the future) perhaps worth discussing. It seems that it isn't known from this study is whether the laminar projection patterns are initially specific or become so upon later refinement (as happens with many projections). It seems possible that cells projecting to deep layers in a limited r/v domain may initially project more widely in the radial domain and then be refined to be in deep layers later? Similarly, the superficially projecting neurons could also be more radially distributed in their termination initially and then refine. This latter possibility for the Layer 1 projecting cells seems perhaps less likely since they project within Layer 1 for a long way and stay in that lamina – but then perhaps there are selective cues for these cells to stay in this lamina that are made by either selective interneurons or CR cells. Again, both of these possibilities seem like interesting follow-up questions.

We are extremely grateful for both the enthusiastic reception, as well as the helpful suggestions raised by the two reviewers. In the revised work, we have addressed the issues raised and trust our work is now suitable for publication. Reviewer one points out that our spatially delineated Cre drivers are too broad to conclude that they do not mask a finer spatial restriction masked within these regions. We whole-heartedly agree with this concern and have revised the manuscript to recognize this caveat accordingly. They also point out that cell death may contribute to the laminar distributions that we observe in the mature cholinergic projections, a point we have also now acknowledged in the revised work. Reviewer two points out that the final precision may reflect some reorganization in the cholinergic fibers due to selective cues emanating from particular populations such as the CR or interneuron populations. We have also revised our work to discuss this interesting possibility.